# Estimation of Metabolic Effects upon Cadmium Exposure during Pregnancy Using Tensor Decomposition

**DOI:** 10.3390/genes13101698

**Published:** 2022-09-22

**Authors:** Yuki Amakura, Y-h. Taguchi

**Affiliations:** 1Graduate School of Science and Engineering, Chuo University, Tokyo 112-8551, Japan; 2Department of Physics, Chuo University, Tokyo 112-8551, Japan

**Keywords:** cadmium exposure, insulin metabolism disruption, tensor decomposition

## Abstract

A simple tensor decomposition model was applied to the liver transcriptome analysis data to elucidate the cause of cadmium-induced gene overexpression. In addition, we estimated the mechanism by which prenatal Cd exposure disrupts insulin metabolism in offspring. Numerous studies have reported on the toxicity of Cd. A liver transcriptome analysis revealed that Cd toxicity induces intracellular oxidative stress and mitochondrial dysfunction via changes in gene expression, which in turn induces endoplasmic reticulum-associated degradation via abnormal protein folding. However, the specific mechanisms underlying these effects remain unknown. In this study, we found that Cd-induced endoplasmic reticulum stress may promote increased expression of tumor necrosis factor-α (TNF-α). Based on the high expression of genes involved in the production of sphingolipids, it was also found that the accumulation of ceramide may induce intracellular oxidative stress through the overproduction of reactive oxygen species. In addition, the high expression of a set of genes involved in the electron transfer system may contribute to oxidative stress. These findings allowed us to identify the mechanisms by which intracellular oxidative stress leads to the phosphorylation of insulin receptor substrate 1, which plays a significant role in the insulin signaling pathway.

## 1. Introduction

Cadmium is a non-essential trace metal found in cells [1]. However, when excessive amounts of Cd enter the body owing to inoculation through food or exposure, it accumulates in the cells. As an electrophilic metal, Cd exerts a negative effect on cellular protein molecules [1]. An adverse effect of excessive Cd accumulation is the inhibition of protein folding. For a protein to fully perform its function, its three-dimensional structure must be maintained properly. Thus, unfolded proteins accumulate in the endoplasmic reticulum, the organelle responsible for proper protein folding. The accumulation of unfolded proteins is known to cause endoplasmic reticulum stress, and excessive stress induces endoplasmic reticulum-associated degradation (ERAD). The ubiquitin-proteasome system is activated in response to ERAD induction to degrade unfolded proteins. Conversely, endoplasmic reticulum stress is alleviated when accumulated unfolded proteins undergo ubiquitination, making them targets for proteasome degradation. However, when stress levels exceed the stress-avoidance capacity of ERAD, cells induce apoptosis. ERAD also promotes ATP-dependent responses. Additionally, insulin signaling is initiated when insulin binds to the insulin receptor (IR) on the cell membrane, which ultimately translocates glucose transporter type 4 (GLUT4) to the cell membrane, allowing glucose to enter the cell. The glycolytic system in cells utilizes glucose taken up by the body to produce ATP. When insulin signaling is inhibited, symptoms, such as elevated blood sugar levels, are observed, which also cause type 2 diabetes.

Some reports have suggested that oxidative stress plays a major role in Cd toxicity. Shaikh et al. reported that Cd toxicity was reduced by administering antioxidants to mice after chronic Cd administration. This suggests that Cd toxicity causes various adverse biological effects via oxidative stress [2].

Several studies have linked Cd toxicity to abnormal insulin metabolism. For example, Takashige et al. reported that mice exposed to high doses of Cd for two weeks might develop abnormal adipocyte differentiation, expansion, and function, leading to insulin resistance, hypertension, and cardiovascular disease [3].

In addition, compelling evidence suggests that prenatal exposure to Cd and other toxic metal contaminants increases the risk of cardiovascular disease and obesity-related morbidity, including type II diabetes, in unborn children [4]. Al-Saleh et al. also reported that prenatal Cd exposure significantly affected birth weight [5]. The data used in this study are based on such reports and are derived from experiments conducted to verify them in mice and to investigate the detailed mechanisms.

The transcriptome analysis results of Jackson et al. (who obtained the data used in this analysis) suggest that Cd exposure induces ERAD and intracellular oxidative stress and mitochondrial dysfunction may cause insulin resistance. Additionally, changes in the expression of genes indicative of lipid abnormalities have been observed in response to Cd exposure [6].

The analysis by Jackson et al. identified differentially expressed genes (DEGs). Specifically, the method was used to determine whether there was a significant difference in expression levels by comparing the ratio of signal values between the genes in the liver transcriptome analysis data of individuals whose mothers were exposed to Cd and those whose mothers were not exposed to Cd at the same point in their lives and of the same gender. In this method, 5789 DEGs were identified for the female postnatal day 42 (PND42) data, which represent more than one-third of the total number of genes in the data identified as DEGs [6].

Thus, although analyses were performed at each time point and the results were compared, no attempts have been made to conduct an integrated analysis of data from all time points or to identify genes that change in response to PND changes.

The linear regression method using data from all time points as explanatory variables (see Section 2.2.1) resulted in the overexpression of 5865 genes. However, to comprehend the mechanism, it is preferable to identify genes that play a decisive role; therefore, the overexpression of more than one-third of all genes is inappropriate.

In this analysis, we used tensor decomposition on the liver transcriptome analysis data of Jackson et al. to identify genes that have some level of time dependence on PND, to identify a smaller number of overexpressed genes relative to the total number of genes, and to estimate which genes play an important role. The study also identified a number of genes that are overexpressed in the PND that are not yet understood. We also aimed to elucidate the mechanisms by which ERAD, intracellular oxidative stress, and mitochondrial dysfunction interact, and to determine how insulin resistance is acquired.

## 2. Materials and Methods

### 2.1. Preparation of the Dataset

“GSE150679” (https://www.ncbi.nlm.nih.gov/geo/query/acc.cgi?acc=GSE150679 (accessed on 16 September 2022)) was downloaded from Gene Expression Omniubus (GEO), a database provided by the National Center for Biotechnology Information (NCBI), USA.

This dataset was obtained by euthanizing offspring born from Cd-exposed female parental mice (exposed to 500 ppb CdCl2 in drinking water from two weeks prior to gestation until birth) and offspring born from control females at PND1, PND21, and PND42, respectively, and performing liver transcriptome analysis. The offspring of Cd-exposed females were similarly exposed to Cd at up to PND 10. For this analysis, they were sequenced with an Illumina (Illumina, Inc. San Diego, CA 92122 USA) HiSeq2500 sequencer. Biological replicates existed for each conditional distinction regarding whether the parent mice were exposed to Cd and when the liver transcriptome analysis was performed. Gene expression levels for each individual in this data were normalized.

The dataset also consisted of three Excel files containing data from PND1, PND21, and PND42.

### 2.2. Model Selection

#### 2.2.1. Significance of Using Tensor Decomposition

In this analysis, we used the Tucker decomposition among the tensor decompositions, and the algorithm was higher-order singular value decomposition (HOSVD [7]). This method is effective for the “Large p Small n problem”. It is useful in situations where the number of specimens is very small relative to the number of genes, as in this analysis, and where multiple genes are selected as variables with some interpretability. Biological experiments, in general, utilize such small numbers of specimens due to ethical considerations and difficulties in obtaining specimens.

As noted above, Jackson et al. identified DEGs based on postnatal days and compared PND1, PND21, and PND42. By transforming the data into a tensor and applying Tucker decomposition, it is possible to identify genes whose expression changes with the passage of PND. This is another reason for our model selection.

In addition, as we will demonstrate in the next section, the simplest imaginable model, the multiple regression model, performs poorly in analyzing such a “large p small n problem”. Tensor decomposition is also a relatively simple model; however, because biological data are typically complex, it is important to select a simple model such as tensor decomposition.

#### 2.2.2. Inappropriate Variable Selection with Multiple Regression Models

The results of using a multiple regression model are used. We set the explanatory variables as the vectors of length 18 and $t_{l}$ and $s_{l}$, where $t_{l}$ is the explanatory variable for PND, and $s_{l}$ is the variable for the presence of Cd exposure.



$$t_{l}=\left(\begin{array}{c} 1\\ 21\\ 42\\ \vdots \\ 1\\ 21\\ 42 \end{array}\right)\;\;s_{l}=\left(\begin{array}{c} 1\\ 1\\ \vdots \\ 1\\ 0\\ 0\\ \vdots \\ 0 \end{array}\right)$$



The target variable was set to $v_{i^{\prime }l}$, and the same value was assigned for a vector of length 18, containing the expression levels of gene $i^{\prime }$. The elements of $v_{i^{\prime }l}$ are the expression levels with Cd exposure at l = 1,…, 9, and subsequent elements are without Cd exposure. Additionally, l=1, l=2, and l=3 were the expression levels of PND = 1, PND = 21, and PND = 42, respectively. Following that, the elements considered to be biological replicates were stored.



$$v_{i^{\prime }l}\;=\;a_{i}+b_{i^{\prime }}t_{l}+c_{i^{\prime }}s_{l}$$



The following multiple regression analysis was repeated 12,795 times for each gene *i* using these variables. Next, for each of the coefficients $b_{i}$ and $c_{i}$ obtained through a multiple regression analysis, we performed the χ-square test and assigned a *p* value. Then, we identified the gene *i* for which p<0.05 was obtained for both coefficients. The number of *i*’s obtained using this method was 5865. However, this method is considered inappropriate because it cannot shortlist the candidate causative genes, since it resulted in the high expression of about 46% of the total genes.

### 2.3. Data Processing

The data were formatted into tensors for use in tensor analysis. First, sex differences were excluded from this analysis owing to insufficient data from conditionally differentiated littermates to formulate a tensor (this is because only the PND42 file has conditional distinctions for sex, and these distinctions were not sufficient in the PND1 and PND21 data to form the tensor).

Since the number of individuals under the same conditions varied between the three files, three datasets of individuals under the same conditions were selected for further analysis, and the columns of the Excel file were rearranged in the following order. The sample ID clearly indicates the order.


PND1:GSM4556351GSM4556352GSM4556353GSM4556347GSM4556348GSM4556349



PND21:GSM4556358GSM4556359GSM4556360GSM4556355GSM4556356GSM4556357



PND42:GSM4556365GSM4556366GSM4556367GSM4556361GSM4556362GSM4556363


The analysis focused on the 12,795 genes whose gene expression levels were consistently obtained in the three files (only PND42 contained data on the expression levels of more genes than the other two).

### 2.4. Structure of the Tensor Used in the Analysis

The variables in each tensor dimension are described after the original data have been formatted. Its structure is illustrated in Figure 1.

The first dimension of the tensor was lined with 12,795 genes.

The second dimension consists of a sample of mice under the same conditions selected from the original data as described previously.

The third dimension represents the passage of time since birth. The above samples were in the order of PND1, PND21, and PND42.

The fourth dimension represents the presence or absence of maternal Cd exposure. In Figure 1, ‘non-control’ represents the Cd-exposed population and ‘control’ represents the unexposed population.

A fourth-order tensor $x_{ijkl}$ was created using these variables. The parameters representing the rank of the matrix in each dimension were set to (*N*, *M*, *K*, *L*) = (12,795, 3, 3, 2).

### 2.5. How to Analysis

#### 2.5.1. Unsupervised Learning Using Tensor Decomposition

In this analysis, we used Tucker decomposition among the tensor decompositions, and the algorithm used was higher order singular value decomposition (HOSVD [7]).

For example, there was a third-order tensor $x_{ijk}$ with ranks *N*, *M*, and *K* in each dimension (the tensor treated in this analysis was a fourth-order tensor, but it was easy to extend the Tucker decomposition from a third- to fourth-order tensor). Tucker decomposition was then performed on this tensor using the orthogonal matrix *U* and decomposed as follows:



$$x_{ijk}\;=\;\sum_{l_{1}=1}^{N}\sum_{l_{2}=1}^{M}\sum_{l_{3}=1}^{K}\;G\left(l_{1},l_{2},l_{3}\right)\;u_{l_{1}i}\;u_{l_{2}j}\;u_{l_{3}k}$$



Tucker decomposition decomposes the original data tensor into the core tensor $G(l_{1},l_{2},l_{3})$ and three singular value matrices $u_{l_{1}i},u_{l_{2}j},u_{l_{3}k}$. These matrices represent the dependencies on variables *i*, *j*, and *k*, respectively.

For example, suppose we want to locate a variable *i* with a large value in the matrix $u_{l_{1}i}$ for analysis. The first step is to select the parameters ${l_{2}}^{\prime }$ and ${l_{3}}^{\prime }$ that correspond to some interest dependence in $u_{l_{2}j}$ and $u_{l_{3}k}$, respectively. For example, if *j* is a variable that represents time variation, it would be preferable that the vector $u_{l_{2}j}$ for the chosen ${l_{2}}^{\prime }$ be straightforward, such as a monotonic increase or decrease vector.

Next, we would substitute the selected ${l_{2}}^{\prime }$ and ${l_{3}}^{\prime }$ into core tensor $G(l_{1},l_{2},l_{3})$. Because each element of the core tensor represents the weight of the product of each vector specified by the parameter $\left(l_{1},l_{2},l_{3}\right)$, the parameter $l_{1}$ exhibits a large absolute value in the resulting vector $G({l_{1}}^{\prime },{l_{2}}^{\prime },{l_{3}}^{\prime })$, which exhibits a large absolute value in the vector $G({l_{1}}^{\prime },{l_{2}}^{\prime },{l_{3}}^{\prime })$, which has a considerable contribution to the original tensor. Based on this idea, we selected ${l_{1}}^{\prime }$.

Next, we obtain the vector $u_{{l_{1}}^{\prime }i}$ by substituting ${l_{1}}^{\prime }$ with $u_{l_{1}i}$. This vector follows the dependency of the chosen parameters ${l_{2}}^{\prime },{l_{3}}^{\prime }$ and has a considerable contribution to the original tensor.

#### 2.5.2. Variable Selection Using χ-Square Test

Finally, the χ-square test was performed using the normalized version of this vector to identify *i* with a particularly high value among the elements of the obtained vector $u_{{l_{1}}^{\prime }i}$, and a *p*-value was assigned.



$$p_{i}\;=\;p_{\chi^{2}}\;\left[>\sum_{{l_{1}}^{\prime }}{\left(\frac{u_{{l_{1}}^{\prime }i}}{\sigma_{{l_{1}}^{\prime }}}\right)}^{2}\right]$$



The above-mentioned objective was achieved by applying a multiple comparison correction to obtain *i* with a *p*-value of less than 0.05.

Figure 2 shows a flowchart of the overall flow of the selection of overexpressed genes in this analysis.

## 3. Results

### 3.1. Identification of Genes with High Expression Levels

When we decomposed the fourth-order tensor used in this analysis in R using HOSVD, we obtained a list of *Z* containing the elements of the core tensor and a list of *U* consisting of the four singular value matrices.

The tensor used was as described previously, where the subscripts *i*, *j*, *k*, and *l* denote the gene, biological replicate, time course, and the presence or absence of maternal Cd-exposure, respectively. The purpose of this analysis was to identify the gene *i*, whose expression level is elevated due to Cd exposure; we investigated whether we could identify a parameter $l_{1}$ in the matrix $u_{l_{1}i}$ that is dependent on Cd exposure, invariant to biological replicates, and sometimes time-dependent. If we could locate a parameter $l_{1}$ in the matrix $u_{l_{1}i}$ that is equivalent to the dependence on Cd exposure, invariance to biological replicates, and some degree of time dependence, we assigned $l_{1}$ to $u_{l_{1}i}$ to obtain the gene *i* of interest. To accomplish this, we first selected $l_{2}$, $l_{3}$, and $l_{4}$.

First, $u_{l_{2}j}$ was extracted from the object *U*. We chose $l_{1}\;=\;1$ because it is a biological replicate; the smaller the individual variation under the identical condition, the better (Figure 3).

Next, we removed $u_{l_{3}k}$. We chose $l_{3}\;=\;2$, which follows a monotonic decrease, because extracting a time dependence that follows a monotonic increase or decrease provides a better interpretation of whether the expression level rises or falls as the days pass from birth (Figure 4).

Next, we removed $u_{l_{4}l}$. Because this study aimed to identify the genes with differential expression in the presence or absence of Cd exposure, $l_{4}\;=\;2$ was selected (Figure 5).

Because $l_{2}\;=\;1$, $l_{3}\;=\;2$, and $l_{4}\;=\;2$ were selected, we sought to identify the genes that exhibit small individual differences under the same conditions, differ with respect to maternal Cd exposure, and exhibit a decreasing expression trend over time.

Next, a vector of length 12,795 was obtained by taking the list of objects *Z* and specifying $G(l_{1},1,2,2)$ and the parameters in the core tensor *G*. The element denoted by $l_{1}$, which has a large absolute value on this vector, is dependent on the parameters $l_{2}\;=\;1$, $l_{3}\;=\;2$, and $l_{4}\;=\;2$, and contributes significantly to the original tensor. Because of the dimensionality of the original tensor, the 19th and subsequent elements on the vector $G(l_{1},1,2,2)$ were very small; therefore, only the values of the elements from 1 to 18 are shown in Figure 6. We then selected $l_{1}\;=\;3,11,14$ with large absolute values as the values of $l_{1}$ to be assigned to $u_{l_{1}i}$. These were substituted into $u_{l_{1}i}$ to generate three vectors.

To identify the genes with relatively high expression levels, we assumed that $u_{3i}$, $u_{11i}$, and $u_{14i}$ followed a normal distribution independently, and assigned *p*-values using the χ-square test. Then, using these three vectors, we executed the p.adjust command.

Additionally, the *p*-value for each gene *i* was adjusted for multiple comparisons using the p.adjust function; then, *i* with a *p*-value of less than 0.05 was selected. The selected gene *i* followed the dependency of interest and corresponded to a gene with a very high expression level. From the total of 12,795 genes, 204 were selected for this analysis.

### 3.2. Checking the Ontology with gProfiler

“gProfiler” (https://biit.cs.ut.ee/gprofiler/(accessed on 16 September 2022)) is a web tool developed by an Estonian research group, the Bioinformatics, Algorithmics and Data Mining Group (BIIT). Ontologies were identified using enrichment analysis on the 204 genes obtained in the above analysis, as shown in Table 1.

Table 1 shows that ontologies include “Lipid metabolism”, “Intracellular signaling”, “Regulation of gene expression”, “Mitochondria”, and “Endoplasmic reticulum”. This could provide a similar interpretation to the results of Jackson et al. In addition, one justification for model selection is the that enrichment analysis has multiple ontology hits. This is because multiple ontology hits are rare, as was the case here when the model was most likely inappropriate and the analysis is failed.

### 3.3. Evaluation of Analysis Results

This analysis used a simple tensor decomposition model to identify 204 genes that were monotonically expressed in response to PND and whose expression levels varied depending on the presence or absence of Cd exposure. The dependence of a monotonic decrease on PND was chosen for clarity of interpretation, indicating that we identified a gene whose expression is maximally elevated when PND1 is overexpressed and then decreases gradually. As previously stated, the analysis by Jackson et al. did not attempt to identify genes that change over time, and we believe our analysis is novel in this respect.

The 204 genes identified represent approximately 1.6% of the total, and compared to the expression variation analysis by Jackson et al. and the multiple regression model presented above, the number of overexpressed genes considerably reduced.

The enrichment analysis also included ontologies that appeared to be highly relevant, which lends credence to this analysis.

## 4. Discussion

### 4.1. Functions of the Identified Gene

This study aimed to identify the mechanism by which insulin signaling is inhibited. We began by determining whether the ERAD-related genes were overexpressed.

As mentioned previously, the ubiquitin-proteasome system promotes protein degradation during ERAD development. The obtained genes included ubiquitin-conjugating enzyme E2 E3 (UBE2E3) with E2 ubiquitin ligase activity [8], autocrine motility factor receptor (AMFR) with E3 ubiquitin ligase activity [9], and others with ubiquitin ligase activity, such as transcription factor E4F1 (E4F1) [10] and tripartite motif-containing protein 32 (TRIM32) [11]. In addition, the COMM domain-containing protein 9 (COMMD9) regulates these activities [12]. Proteasome-related genes include PSMC2, which encodes a subunit of the 26s proteasome [13], and proteasome activator complex subunit 1 (PSME1), which is part of the proteasome activator complex PA28 the 11S regulator, known as PA28 [14]. The elevated expression of multiple genes involved in the ubiquitin–proteasome system may be responsible for the development of ERAD.

The genes involved in the nuclear factor-κ B (NF-κB) pathway, which plays a central role in the immune response, were TANK-binding kinase 1 (TBK1), which indirectly interacts with NF-κB [15]; caspase recruitment domain-containing protein 11 (CARD11), which is an activator [16]; and COMMD9, which is a negative regulator [17], suggesting that NF-κB may be highly expressed. It is activated by the cytokine TNF-α, and TNF-α may be highly expressed due to the high expression of NFAT activating protein with ITAM motif 1 (NFAM1), which encodes a receptor that activates the promoter of TNF-α [18]. Previous studies have shown a positive correlation between TNF-α expression and development [19], and the results of this study are consistent with this finding. Given that ERAD is ATP-dependent, we verified whether any genes that upregulate ATP production were highly expressed and identified ATP synthase membrane subunit c locus 1 (ATP5G1), which encodes a subunit of the enzyme that catalyzes ATP synthesis.

The genes involved in the electron transfer system were NADH ubiquinone oxidoreductase complex assembly factor 7 (NDUFAF7), which is involved in stabilizing the assembly of electron-transfer complex enzymes [20], and electron transfer flavoprotein subunit beta lysine methyltransferase (ETFBKMT), which negatively regulates the electron-transfer system function [21]. The enzyme dihydrolipoamide s-acetyltransferase (DLAT), which links the electron-transfer system to the glycolytic system [22], was also highly expressed. The metabolites of glycolysis and fatty acid metabolism are transported to the electron transport system to promote ATP production in mitochondria. The genes involved in fatty acid metabolism were glycosylphosphatidylinositol anchored high density lipoprotein binding protein 1 (GPIHBP1), which promotes fatty acid degradation [23], and peroxisome proliferative activated receptor alpha (PPARA), which encodes a growth factor-activated receptor for peroxisomesn [24], the site of fatty acid metabolism. Considering that these metabolites are transferred to the electron transfer system, the activation of the electron transfer system may be responsible for the high expression of genes involved in glycolysis and fatty acid metabolism. The activation of the electron transfer system stimulates ATP production directly.

Additionally, genes involved in sphingolipid biosynthesis, serine palmitoyltransferase, long chain base subunit 2 (SPTLC2) [25] and sterile alpha motif domain containing 8 (SAMD8) [26], were highly expressed. This suggests that cells may over-synthesize sphingolipids.

### 4.2. Elucidation of the Mechanism of Insulin Metabolism Inhibition from the Obtained Genes

Cd exposure of the mother also exposes the fetus to Cd [27]. Excessive cellular Cd uptake inhibits protein folding, which leads to ERAD development. This analysis suggests that the high expression of genes involved in ubiquitin–proteasome system glycolysis, fatty acid metabolism, and electron transfer systems promote ATP synthesis, which may assist ERAD development. The positive correlation between the expression levels of ERAD and TNF-α, and the high expression of genes related to TNF-α and NF-κB, suggest that TNF-α may be over-secreted in the cells. TNF-α is known to activate ASMase (oxidized sphingomyelinase), and ASMase promotes the production of sphingolipids. In this analysis, a high expression of genes involved in sphingolipid synthesis was also observed, suggesting that sphingolipid accumulation may occur. The accumulation of ceramide, a sphingolipid, leads to the overproduction of reactive oxygen species (ROS). Additionally, ROS are produced during the electron transfer system reaction. ROS-mediated oxidative stress activates the apoptosis signal-regulating kinase 1 (ASK1) [28]. Phosphatidylserine decarboxylase proenzyme(PISD) was one of the highly expressed genes identified in this study. It is also involved in the activation of ASK1. ASK1 activates c-Jun N-terminal kinases (JNK), and JNK inactivates it by phosphorylating 307 serine residues of IRS1, which are essential for insulin signaling. Consequently, a pathway for the development of insulin resistance due to oxidative stress via TNF-α-induced ceramide accumulation can be inferred from this analysis. However, from this analysis, we could not elucidate the mechanism underlying the positive correlation between ERAD and the high expression of TNF-α.

In addition to the genes mentioned above, there were also hits for ontologies such as “Regulation of gene expression” and “RNA metabolism”, as shown in Table 1. This suggests that gene expression levels were altered, but the specific mechanism could not be determined in this study.

We also found a number of genes related to apoptosis, which is a last resort for stress avoidance in ERAD development, but it is unclear how apoptosis is actually related to the pathway of insulin resistance acquisition, which we speculated in this study.

In future studies, we plan to focus on determining how these factors are involved and, in particular, what causes and influences gene expression changes.

## 5. Conclusions

Based on a study reporting that Cd exposure during pregnancy disrupts insulin metabolism in offspring, 204 overexpressed genes were identified using tensor decomposition with liver transcriptome analysis data to elucidate the cause of this biological phenomenon.

Included among the 204 genes were those involved in ERAD development and the ubiquitin-proteasome system, a stress avoidance pathway, implying that Cd toxicity inhibits protein folding. A group of genes involved in sphingolipid production and the electron transfer system was also highly expressed, indicating excessive intracellular oxidative stress. The ERAD and intracellular oxidative stress results were consistent with those of Jackson et al. [6].

It has been suggested that TNF-α may serve as a link between ERAD and high sphingolipid expression caused by Cd toxicity. The high expression of genes encoding the promoter of TNF-α and related to NF-κB supported this hypothesis. However, this analysis was not able to determine the precise mechanism of the association between ERAD and TNF-α. Based on the association between TNF-α and sphingolipids, we could deduce a pathway by which TNF-α activates ASMase.

An intracellular oxidative stress-based pathway was hypothesized regarding the onset of insulin resistance.

The effects of cadmium on changes in gene expression at the transcriptional level must continue to be investigated.

## Figures and Tables

**Figure 1 genes-13-01698-f001:**
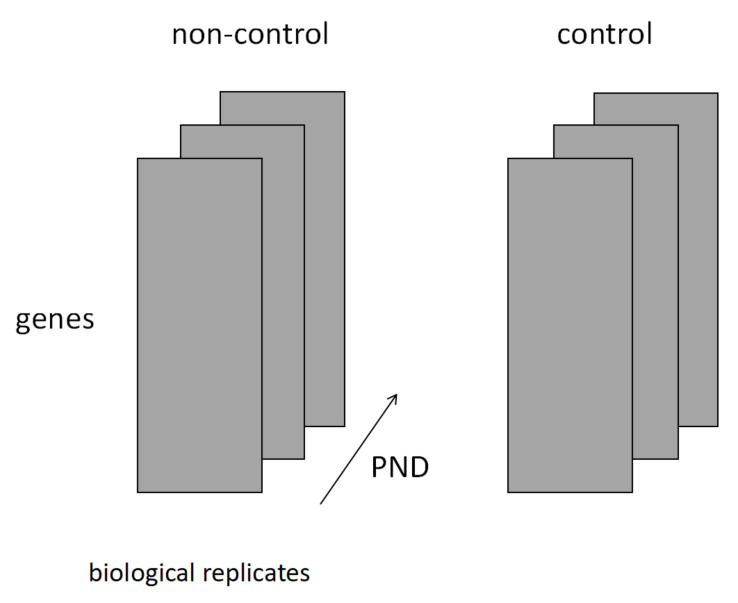
The following figure depicts the four order tensors used in the analysis. The first dimension describes the genes, biological replicates are provided in the second dimension, time course is expressed as postnatal day (PND) in the third dimension, and presence or absence of Cd exposure is indicated in the fourth dimension (non-control refers to offspring born from Cd-exposed mothers, while control refers to offspring born from Cd-unexposed mothers.). Let *N*, *M*, *K*, and *L* denote the ranks of each dimension, (*N*, *M*, *K*, *L*) = (12,795, 3, 3, 2).

**Figure 2 genes-13-01698-f002:**
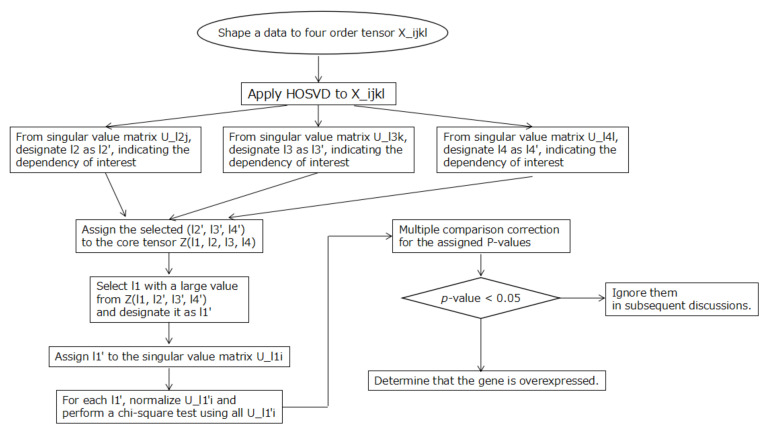
The entire flow of this analysis is represented in the flowchart diagram.

**Figure 3 genes-13-01698-f003:**
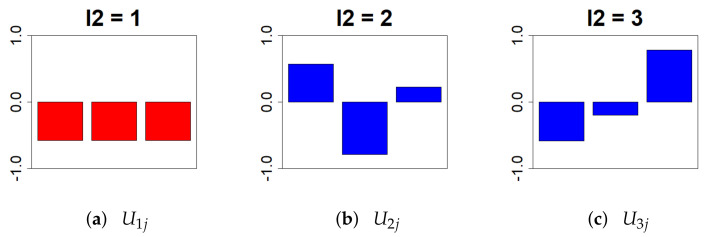
This bar chart represents the values of the singular value matrix $U_{l_{2}j}$ on the column vector. (First column: (**a**), second column: (**b**), and third column: (**c**)). For the analysis, there should be no variation in gene expression between individuals under identical conditions. Therefore, we chose $l_{2}$ = 1.

**Figure 4 genes-13-01698-f004:**
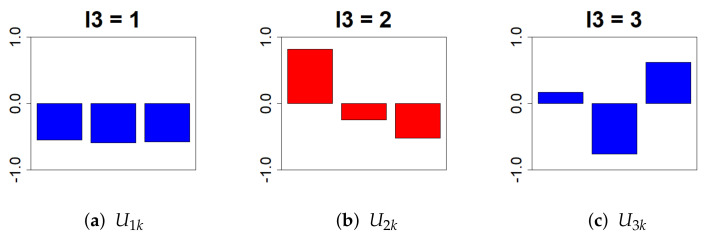
This bar chart represents the values of the singular value matrix $U_{l_{3}k}$ on the column vector. (First column: (**a**), second column: (**b**), and third column: (**c**)). The objective is to isolate genes whose expression varies time-dependently due to Cd exposure. Therefore, we selected $l_{3}$ = 2, which shows a monotonic decrease relative to PND.

**Figure 5 genes-13-01698-f005:**
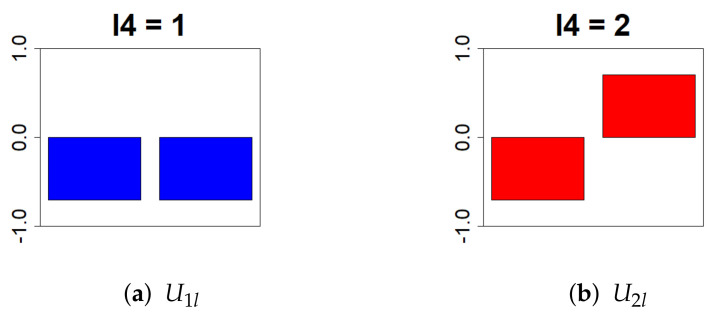
This bar chart represents the values of the singular value matrix $U_{l_{4}l}$ on the column vector. (First column: (**a**), second column: (**b**)). The objective is to isolate genes that show differences in gene expression levels due to Cd exposure. Therefore, we selected $l_{4}$ = 2, which has distinct positive and negative values.

**Figure 6 genes-13-01698-f006:**
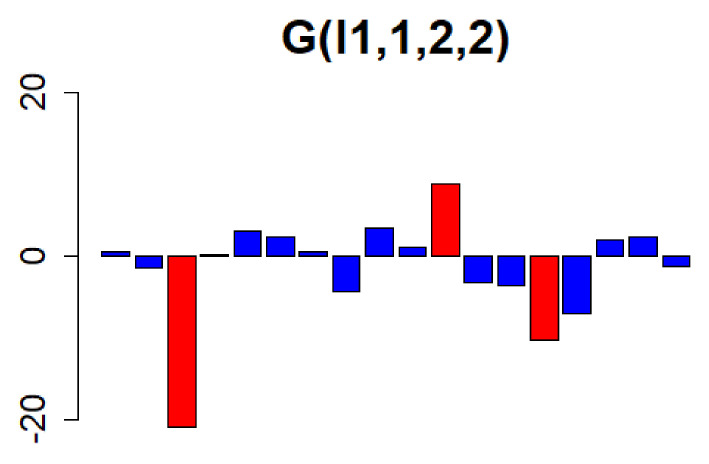
The values of the vectors with ($l_{2}$ = 1, $l_{3}$ = 2, and $l_{4}$ = 2) on the core tensor, from the first element to the eighteenth element, are represented using a bar chart. The top three with the highest values ($l_{1}$ = 3, 11, 14) were selected. Owing to the rank of the tensor before decomposition, the nineteenth and subsequent elements are omitted because their values are very small.

**Table 1 genes-13-01698-t001:** Enrichment analysis results using gProfiler. The ontology of hits, database IDs, and *p*-values for each gene were recorded for the 204-gene enrichment analysis.

Database	Term Name	Term ID	*p*-Value
KEGG	Metabolic pathway	KEGG01100	1.661 × $10^{-6}$
KEGG	Activation of platelets	KEGG04611	2.294 × $10^{-2}$
REAC	Lipid metabolism	R-HSA-556833	1.107 × $10^{-2}$
GO BP	Ionic transport	GO0006811	2.741 × $10^{-8}$
GO BP	Cell adhesion	GO0007155	4.702 × $10^{-6}$
GO BP	Intracellular signaling	GO0035556	4.207 × $10^{-6}$
GO BP	Cellular protein modification process	GO0036211	5.519 × $10^{-9}$
GO BP	Regulation of gene expression	GO0010468	1.241 × $10^{-8}$
GO BP	RNA metabolism	GO0016070	9.895 × $10^{-11}$
GO CC	Mitochondria	GO0005739	1.990 × $10^{-4}$
GO CC	Endoplasmic reticulum	GO0005783	1.990 × $10^{-3}$

## Data Availability

Gene Expression Omnibus (GEO), a database provided by the National Center for Biotechnology Information (NCBI), USA at https://www.ncbi.nlm.nih.gov/geo/query/acc.cgi?acc=GSE150679 (accessed on 16 September 2022).

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
