# Peer review of "Estimation of Metabolic Effects upon Cadmium Exposure during Pregnancy Using Tensor Decomposition"

_genes, 2022, doi:10.3390/genes13101698_

Round 1

Reviewer 1 Report

Comments to Authors

The paper predicts the cadmium exposure on pregnant metabolism using tensor decomposition. Below are some comments and suggestion for improving the quality of the manuscript.

General comments

1.      The manuscript needs to be improved in term of English language.

2.      The authors failed to convince the reader the rationale behind their work. The authors must address why it significant to have a model.

3.      The structure of the paper need to be improved. For example, section 2.3 contains five sub-sections, each contains only one or two lines. The analysis method can be explained using a visual figure and described in one or two paragraphs.

Abstract

Although the abstract gives a good overview of the methodology followed. It does not give sufficient details about the study results.

Introduction

The motivation of the study and the research gaps derived from the literature review is not clearly stated.

Related work

1.      A summary of the previous study is missing

2.      The limitations/gaps of the previous studies are missing.

Proposed Method

1.      The selection of methods should be justified, for example, why tensor composition is an appropriate method for the proposed solution.

2.      It is not clear how the authors evaluate the proposed model.

Results

1.      Figures 1-4 should be given complete captions that tell the reader what these figures show.

2.      I suggest to summarize the results found by listing the major findings.

3.      To evaluate the proposed model, the authors have to compare the results with the state of the models.

Conclusion

The implication of the work should be discussed

References

References are outdated, no single reference published in 2022 or 2021. Most of them are before 2015.

Author Response

Dear Reviewer

Thank you very much for your advice on my recent paper submission.

We also thank you for your flexibility in responding to our request for an extension of the revision period.

We have finished revising the points raised by the reviewers and are pleased to submit a revised manuscript. I will also write in this letter how I have improved the revisions.

Revised sections are in red in the manuscript. Material deleted since the last submission is crossed out with a red line. Also attached is a copy of the manuscript with all corrections, etc., in black ink. (Please note that corrections made after proofreading are not marked in red.)

Please check the manuscript.

  • Points raised by Reviewer 1

General comments

  1. The manuscript needs to be improved in term of English language.

・ Revision details

The manuscript was proofread prior to initial submission. The manuscript was edited again for this revision.

  1. The authors failed to convince the reader the rationale behind their work. The authors must address why it significant to have a model.

・ Revision details

Because of the complexity of biological data, it makes sense to assume a simple model of tensor decomposition itself. I wrote this newly in "2.2.1 Significance of using tensor decomposition" in the manuscript.

  1. The structure of the paper need to be improved. For example, section 2.3 contains five sub-sections, each contains only one or two lines. The analysis method can be explained using a visual figure and described in one or two paragraphs.

・ Revision details

The subsection corresponding to "2.3 Tensor used in analysis" in the previous submission was rewritten as "2.4 Structure of the tensor used in the analysis" and the structure of the subsection pointed out was improved. In addition, a figure was created and inserted to provide a visual image of the tensor.

Abstract

Although the abstract gives a good overview of the methodology followed. It does not give sufficient details about the study results.

・ Revision details

Overall changes were made. In particular, the content of the results section was enhanced.

Introduction

The motivation of the study and the research gaps derived from the literature review is not clearly stated.

・ Revision details

A note was added regarding the motivation for the study and the research gap derived from the literature review. Several references were added accordingly.

Related work

  1. A summary of the previous study is missing
  2. The limitations/gaps of the previous studies are missing.

・ Revision details

Several relevant previous studies were added in the "Introduction" section. In addition, we described the novelty of this study based on them.

Proposed Method

  1. The selection of methods should be justified, for example, why tensor composition is an appropriate method for the proposed solution.

・ Revision details

A new subsection "2.2 Model Selection" was added to explain the justification for the choice of methodology. Also, the section "5. Comparison with other methods" from the previous submission was moved to this section and the sub-sub-section was renamed "2.2.2 Inappropriate variable selection with multiple regression models".

  1. It is not clear how the authors evaluate the proposed model.

・ Revision details

In actually proceeding with the analysis, it is rare that the enrichment analysis will produce more than one significant one if the analysis is not successful. This and the overall analysis evaluation are summarized in a new subsection, “3.3Evaluation of Analysis Results”.

Results

  1. Figures 1-4 should be given complete captions that tell the reader what these figures show.

・ Revision details

Corrected captions for all figures, including newly added figures.

  1. I suggest to summarize the results found by listing the major findings.

・ Revision details

I wrote in "3.3Evaluation of Analysis Results."

  1. To evaluate the proposed model, the authors have to compare the results with the state of the models.

・ Revision details

The results of the previous studies and the comparison with multiple regression models were compared with the contents of "Introduction" and "2.2.2 Inappropriate variable selection with multiple regression models" and described in "3.3. Evaluation of Analysis Results".

Conclusion

The implication of the work should be discussed

・ Revision details

The significance of this study is discussed in the Introduction.

References

References are outdated, no single reference published in 2022 or 2021. Most of them are before 2015.

・ Revision details

We do not think it is necessary to correct this point. This is because many of the references are to the mechanism of action of genes, and it is difficult to imagine that there is an up-to-date version of those references.

This is all about the corrections and improvements to the points raised.

Reviewer 2 Report

The idea behind this manuscript is quite interesting and could be considered for publication if the manuscript is well-written. At the moment, this version cannot be accepted for publication citing the following reasons:

1. Several acronyms were not defined at first use, including ATP, GLUT4, TNF, IRS1, etc. It is very important to define acronyms even though they are well-known and somehow provide sufficient explanation where necessary. 

2. The introduction section is very poor and not easy to understand; the ideas are provided rather haphazardly. The authors need to provide an informative introduction, present details about the study, including the "why" and "how", and provide sufficient citations and comparisons with other published works.  

3. In Section 2, the dataset is not discussed properly. The information provided is not sufficient, and there is no citation pointing to where the dataset can be found. 

4. Section 2.4. titled "How to analysis" is rather confusing. 

5. Section 2.4.1. how the Higher-Order Singular Value Decomposition was used as the algorithm is not sufficiently described. 

6. There is not sufficient information on how the Chi-square was used for feature selection. 

7. The results are too rudimentary and cannot be considered sufficient for publication. 

8. There is no Conclusion section. Why was this not excluded?

Author Response

Dear Reviewer

Thank you very much for your advice on my recent paper submission.

We also thank you for your flexibility in responding to our request for an extension of the revision period.

We have finished revising the points raised by the reviewers and are pleased to submit a revised manuscript. I will also write in this letter how I have improved the revisions.

Revised sections are in red in the manuscript. Material deleted since the last submission is crossed out with a red line. Also attached is a copy of the manuscript with all corrections, etc., in black ink. (Please note that corrections made after proofreading are not marked in red.)

Please check the manuscript.

Points raised by Reviewer 2

1.Several acronyms were not defined at first use, including ATP, GLUT4, TNF, IRS1, etc. It is very important to define acronyms even though they are well-known and somehow provide sufficient explanation where necessary.

・ Revision details

All abbreviations for genes, metabolic pathways, etc. were written by their formal names at the first use.

2.The introduction section is very poor and not easy to understand; the ideas are provided rather haphazardly. The authors need to provide an informative introduction, present details about the study, including the "why" and "how", and provide sufficient citations and comparisons with other published works. 

・ Revision details

The comparison with the results of other papers and with the results of our own multiple regression model are described in "3.3 Evaluation of Analysis Results".

3.In Section 2, the dataset is not discussed properly. The information provided is not sufficient, and there is no citation pointing to where the dataset can be found.

・ Revision details

The content of "2.1 Preparation of the dataset" was enhanced with respect to the description of the dataset. Footnotes were also included at the bottom of the page regarding the source of the data citations.

  1. Section 2.4. titled "How to analysis" is rather confusing.

・ Revision details

A flowchart diagram of the entire analysis flow was prepared and attached.

  1. Section 2.4.1. how the Higher-Order Singular Value Decomposition was used as the algorithm is not sufficiently described.

・ Revision details

The flowchart diagram shows how Higher-Order Singular Value Decomposition is used in the overall flow.

6.There is not sufficient information on how the Chi-square was used for feature selection.

・ Revision details

The contents of the remarks are described in "2.5.2 Variable selection using χ-square test".

The flowchart also shows how the χ-square test is used in the overall process.

  1. The results are too rudimentary and cannot be considered sufficient for publication.

・ Revision details

The content of this analysis differs from that of previous studies in that it treats the data in an integrated manner and attempts to identify a smaller number of genes with a definitive role. We also believe that we actually obtained a small number of genes as a result and were able to infer a mechanism via overexpression of TNF-alpha that has not been mentioned in previous studies. These are the significance of this study.

8.There is no Conclusion section. Why was this not excluded?

・ Revision details

"5. Conclusion" was added to summarize the overall results.

This is all about the corrections and improvements to the points raised.

Round 2

Reviewer 1 Report

All comments related to manuscript contents have been addressed but I recommend performing proofreading and correcting formatting issues.